# Elderly Patients Managed Non-Operatively with Abscesses of the Anorectal Region Have Five Times Higher Rate of Mortality Compared to Non-Elderly

**DOI:** 10.3390/ijerph20075387

**Published:** 2023-04-04

**Authors:** Alexander Ladinsky, Abbas Smiley, Rifat Latifi

**Affiliations:** 1School of Medicine, New York Medical College, Valhalla, NY 10595, USA; 2Westchester Medical Center, Valhalla, NY 10595, USA; 3Department of Surgery, College of Medicine-Tucson, University of Arizona, Tucson, AZ 85721, USA

**Keywords:** abscess, anorectal, mortality, age, operation, HLOS, elderly, adult

## Abstract

This study’s purpose was to investigate risk factors for mortality from anorectal abscesses through a more comprehensive examination. This was a retrospective study that evaluated National Inpatient Sample patient data of adult and elderly patients emergently admitted with a primary diagnosis of anorectal abscess. Data was stratified by variables of interest and examined through statistical analysis, including backward logistic regression modelling. Roughly 40,000 adult patients and nearly 7000 elderly patients were admitted emergently with a primary diagnosis of abscess in anorectal regions. The mean age of adult male patients was 43 years while elderly male patients were, on average, 73 years old. Both adult males (69.0%) and elderly males (63.9%) were more frequently seen in the hospital for anorectal abscess compared to females. Mortality rates were lower in adult patients as only 0.2% (n = 62) of adult patients and 1.0% (n = 73) of elderly patients died in the hospital. Age increased the odds of mortality (OR = 1.03; 95% CI: 1.02–1.04, *p* < 0.001) as did hospital length of stay (OR = 1.02; 95% CI: 1.01–1.03, *p* < 0.001). Surgical procedure decreased the odds of mortality by more than 50% (OR = 0.49; 95% CI: 0.33–0.71, *p* < 0.001). Risk factors for mortality from anorectal abscess included age and non-operative management, which leads to prolonged hospital length of stay. Surgical management of anorectal abscesses offered protective benefits.

## 1. Introduction

Abscesses of the perirectal region are common causes for surgical consultation that often require invasive procedures [1]. Anorectal abscesses are known to be infections of the anal gland which spread into proximal locales and may often result in fistulas. These fistulas can be problematic and result in incontinence or even cancer if long-lasting [2]. Additionally, although rare, anorectal abscesses that do not resolve with medications and other conservative treatments can sometimes in actuality be lymphomas [3]. The estimated incidence of anorectal abscess is roughly 16 to 20 for every 100,000 patients [4,5]. In the US, the incidence of anorectal abscess is estimated at roughly 68,000 to 96,000 patients per year [6,7]. In the UK, there are roughly 14,000 to 20,000 cases and 12,500 surgeries related to anorectal abscesses each year [8]. Fistulas are frequent following anorectal abscess treatment, with an incidence between 26–37% [2]. 

The etiology of anorectal abscesses consists of an infection of the anal gland. When the infection is between the internal sphincter and external sphincter, it is considered perianal; an infection located solely in the external sphincter is considered ischiorectal; an infection coming anteriorly through the rectal wall is an intermuscular abscess; an infection coming above through the levators is a supralevator abscess; and an infection located in the ischiorectal fossae is known as a horseshoe abscess [2]. Common risk factors for anorectal abscesses include male sex, smoking, diabetes, HIV, or other immunosuppressive circumstances that impede wound healing [9]. Interestingly, lower socioeconomic status was found to be associated with fistula formation after incision and drainage for the abscess [10]. Being able to properly diagnose and then subsequently classify the anorectal abscess is quite important, as finding the internal orifice of anorectal abscesses significantly decreases rates of recurrence of both fistulas and abscesses [11]. A correct early surgical treatment protocol and proper wound-care monitoring for abscesses after surgery is imperative not only for preventing recurrence, but also for preventing mortality. If the abscess is not diagnosed and corrected in a timely manner, the abscess can potentially lead to perianal sepsis or necrotizing soft tissue infection (NSTI) of the perineum and multiple organ system failure, requiring major operation and diversion of the fecal stream [9]. Early and aggressive surgical treatment of NSTI has been proven to be the most important factor in prognosis of these infections [12,13], which may have mortality rates approaching 50% [9]. Thus, clearly, a significant risk factor of mortality from anal abscess is delayed or improper treatment of the abscess. The other known major risk factors for mortality include an immunocompromised status, Crohn’s Disease, and advanced rectal cancer [9]. 

Recurrence rates and mortality rates increase significantly with delayed treatment even in a healthy population; this makes swift diagnosis, proper treatment, and careful monitoring imperative to help decrease the risks from anorectal abscess. The purpose of this study was to analyze patients who were admitted emergently to the hospital with a primary diagnosis of perirectal abscess. This was done on a large scale and over a 10-year period, with the goal to observe those patients’ characteristics and also their relationship to mortality. This paper’s research is especially important because, while there is copious research on risk factors leading to recurrence of anorectal abscess and fistula, there is a dearth of research exploring various risk factors for mortality in patients with anorectal abscesses. 

## 2. Materials and Methods

This research made use of the National Inpatient Sample (NIS) databases. The database was created by AHRQ (the Agency for Healthcare Research and Quality), which has been frequently used nationally as a public data source for analysis of variegated types and qualities of patient care and their associated results. This has allowed us to perform a comprehensive and holistic path to research diseases, a path to the ideal way to treat and care for patients with those diseases, and ultimately a path to find how patients respond to both the diseases and treatments [13,14,15,16,17,18,19,20,21,22,23,24,25]. The NIS database includes certain weighting when constructing its sample of discharges and it excludes long-term acute care facilities and rehabilitation centers. These particular categorizations allowed the NIS database to parse out predictors of nation-wide hospitalizations. This paper consists of retrospective research encompassing patients with a primary diagnosis of an abscess in the anal or rectal region (ICD-9 code 566). These patients were emergently admitted from 2004 to 2014 (2005–2014 in the regression), and were part of the National Inpatient Sample database. The data were organized and subsequently examined in terms of age, with adults classified as ages 18–64 and elderly patients classified as ages 65 and older. The data was then further stratified into sex categories, patient mortality, and operation status (operation or no operation). The variables of interest were age, race (White, Black, Hispanic, Asian/Pacific Islander, or other), income quartile, insurance type (private, Medicare, Medicaid, self-pay, no-charge, or other), hospital location (rural, urban teaching, or urban non-teaching), comorbidity (AIDS, alcohol abuse, deficiency anemias, rheumatoid arthritis, chronic blood loss, CHF, chronic pulmonary disease, coagulopathy, depression, uncomplicated diabetes, chronic diabetes, drug abuse, hypertension, hypothyroidism, liver disease, lymphoma, fluid or electrolyte disorders, metastatic cancer, other neurological disorders, obesity, paralysis, peripheral vascular disorders, psychoses, pulmonary circulation disorders, renal failure, solid tumor, peptic ulcer, valvular disease, or weight loss), invasive diagnostic procedures, surgical procedures, medical or surgical complications, reoperation, deceased status, time to invasive diagnostic procedure, time to surgical procedure, hospital length of stay, and total charges in American dollars. 

The modified frailty index in our study was scored from 0 to 5, with 5 being the frailest, and 0, not being frail. This modified frailty index included 5 variables: congestive heart failure, chronic obstructive pulmonary disease, hypertension, diabetes mellitus, and a patient’s functional status. The index was then calculated via summation of one point of the following disease histories. A history of diabetes was included in the index if patients had either diabetes with chronic complications or uncomplicated diabetes. Congestive heart failure was marked to be included in the index for a patient if that patient had this comorbidity. Patient history of hypertension was included if hypertension was a comorbidity for the respective patient. Patients were marked for having a history of COPD if chronic obstructive pulmonary disease was noted for that respective patient. The NIS data did not explicitly include ‘functional health status’, so it was extrapolated from other comorbidities in the data set; patients with tumors, metastatic cancer, kidney failure, coagulopathy, lymphoma, paralysis, or weight loss were assumed to have some level of dependence and were thus marked as functionally dependent. 

### Statistical Analysis

The data were organized into tables (Table 1, Table 2 and Table 3) in which the correlated number and percentage or mean and standard deviation were presented. These tables were then analyzed using the chi-squared test for categorical variables and a *t*-test for continuous variables. Risk factors for mortality were analyzed through backward elimination multivariable logistic regression analysis (Table 4). The risk factors were analyzed and presented as odds ratios with 95% confidence intervals, and *p* values below 0.05 were considered significant. Characteristics considered for adjustment in the model were age, surgical procedure, hospital length of stay in days, invasive procedure, respiratory diseases, cardiac diseases, liver diseases, genitourinary system diseases, platelet and white blood cell diseases, trauma, burns and poisons, neoplasms, and neurological diseases. Table 5 was analyzed in the same manner as Table 1, Table 2 and Table 3. We used SPSS version 24 (SPSS Inc., Chicago, IL, USA) and R software (Foundation of Statistical Computing, Vienna, Austria). 

## 3. Results

### 3.1. Gender Differences

A total of 40,046 adult patients (age 18–64) were admitted emergently with a primary diagnosis of abscess in anal or rectal regions from 2004 to 2014. Of these, 27,646 (69.0%) were male while 12,400 (31.0%) were female, with a mean (SD) age of 43.05 (11.69) and 41.47 (12.28), respectively. On the other hand, there were 6982 elderly patients (age 65+) of which 4463 (63.9%) were male while 2519 (36.1%) were female, with a mean (SD) age of 73.42 (6.96) and 76.08 (8.11), respectively. For adults, most patients were White, funded by private insurance, and admitted to urban teaching hospitals. For elderly patients, most were White and largely funded by Medicare (Table 1). 

Major comorbidities for adults included hypertension, uncomplicated diabetes, obesity, and fluid/electrolyte disorders. Adult males manifested higher rates of alcohol abuse, AIDS, drug abuse, liver disease, and paralysis, while adult females had higher rates of deficiency anemias due to chronic blood loss, rheumatoid arthritis, chronic pulmonary disease, depression, diabetes, hypothyroidism, fluid or electrolyte disorders, neurological disorders, obesity, and psychoses. Major comorbidities for elderly patients included hypertension, uncomplicated diabetes, fluid or electrolyte disorders, and deficiency anemias. Elderly males presented with higher rates of alcohol abuse, coagulopathy, lymphoma, metastatic cancer, and peripheral vascular disorders, while elderly females manifested higher rates of deficiency anemias, rheumatoid arthritis, depression, hypertension, hypothyroidism, fluid or electrolyte disorders, obesity, psychoses, and weight loss. 

Adult females had longer hospital length of stay (HLOS), longer time to invasive diagnostic procedures, and longer time to surgical procedures than adult males. Adult females also were charged more money than adult males. Elderly females also had a longer HLOS than elderly males, but there was no significant difference between genders in terms of time to procedures and hospital charges. Elderly male patients had higher rates of surgical and invasive procedures than elderly female patients, but there was no significant gender difference in adult patients (Table 1). 

### 3.2. Operative vs. Non-Operative Treatment

There were 34,137 (84.7%) adult patients admitted who underwent an operation in comparison to 6169 (15.3%) admitted adult patients who did not have an operation. A total of 5370 (76.9%) of the elderly patients underwent surgical procedure in comparison to 1616 (23.1%) elderly patients who did not have an operation (Table 2). For adults, the mean age of the group of patients who had an operation was 1.19 years younger than the group of patients who had no operation. For elderly patients, the mean age of the group of patients who had an operation was 1.40 years younger than the group of patients who had no operation (Table 2). For adults, in both the operative and non-operative groups, most patients were male, White, in income quartile 1, funded by private insurance, and admitted to urban teaching hospitals. For the elderly, in both the operative and non-operative groups, most patients were male, White, in income quartile 1, funded by Medicare, and admitted to urban non-teaching hospitals. In both adults and elderly patients, the mortality rate of operated patients was significantly lower than those managed non-operatively. For adults, those who had an operation had lower rates of incidence for most comorbidities. For elderly patients, there were fewer significant differences in comorbidity frequencies between those operated and not operated on; when there were significant differences, those who had an operation also had lower incidences of comorbidities, except for hypertension. Additionally, both adult and elderly patients who were operated on had higher rates of invasive diagnostic procedures, shorter hospital length of stay, and shorter time to invasive diagnostic procedure compared to those who did not have an operation. There was no significant difference in costs for adult patients regardless of whether they underwent operation, but among elderly patients, those who had an operation had higher total charges from hospitals than those who did not have an operation (Table 2).

### 3.3. Mortality and Age

Only 0.2% (n = 62) of adult patients died in the hospital, while 1.0% (n = 73) of elderly patients died in the hospital. Thus, the mortality rate in elderly patients was roughly five times the rate in nonelderly patients. While adult patients who survived as well as those who did not survive predominantly had private insurance, deceased patients were more likely to have used Medicare and Medicaid, while patients who survived were more likely to have used private insurance or self-pay (Table 3). Additionally, adult deceased patients had a larger percentage that were at urban teaching hospitals compared to those who survived. There was no significant difference between elderly patients who survived or died in terms of insurance or hospital location (Table 3). The deceased adults were roughly 9 years older than the adults that survived, while the deceased elderly patients were roughly 5 years older than the elderly patients who survived. The deceased elderly patients had a significantly longer time to invasive diagnostic and surgical procedure than those who survived; deceased adults had a significantly longer time to surgical procedure, but no significant difference in time to invasive diagnostic procedure compared to those adults who survived. Both deceased elderly and deceased adult patients had a longer hospital length of stay than their counterparts who survived: the elderly who died stayed over 6 days longer (on average) than those who survived, and adult patients who died stayed over 11 days longer (on average) than those who survived (Table 3). Similarly, total hospital charges were larger for both adult and elderly deceased patients than patients who survived.

There were also differences in terms of comorbidities exhibited by deceased patients and those who survived. For adults, those who died in the hospital were more likely to exhibit alcohol abuse, deficiency anemias, congestive heart failure, coagulopathy, hypertension, liver disease, lymphoma, fluid/electrolyte disorders, metastatic cancers, neurological disorders, peripheral vascular disorders, renal failure, and weight loss. For elderly patients who died in the hospital, they were more likely to present with congestive heart failure, chronic pulmonary disease, coagulopathy, fluid or electrolyte disorders, metastatic cancer, neurological disorders, peripheral vascular disorders, pulmonary circulation disorders, renal failure, and weight loss. This is similar to the study by Latifi et al. that demonstrated that presence of comorbid conditions such as alcohol abuse, peripheral vascular disease, diabetes, obesity, and hypothyroidism were associated with increased mortality in patients with NSTI, which is often associated with death in anorectal abscess patients [13].

Both elderly and adult deceased patients also exhibited higher rates of reoperation, medical or surgical complication diagnosis, and invasive diagnostic procedures; both elderly and adult patients who survived exhibited higher rates of surgical or invasive procedures. Elderly patients who survived had higher rates of uncomplicated diabetes (Table 3).

### 3.4. Risk Factors of Mortality

The backward logistic regression model, with mortality as the dependent variable, was created to assess associations of mortality and different risk factors for patients. Variables used to adjust the model included age, surgical procedure, invasive procedure, respiratory disease, cardiac disease, liver disease, genitourinary disease, platelet and white blood cell disease, trauma, burns and poisons, neoplasms, and neurological disease (Table 4). Management of patients through operation decreased the odds of mortality by 51%. Each additional year aged (in our patient sample of adults and elderly) increased the odds of mortality by 3%. Each additional day of hospital stay increased the odds of mortality by 2%. Having a surgery was a protective factor, however, as non-operative management doubled the odds of mortality. As we can see, trauma, burns, and poisons increased the odds of mortality more than 7-fold. Respiratory disease increased the odds of mortality more than 4-fold, and both cardiac and liver diseases increased the chances of mortality by greater than 3 times. Genitourinary system diseases increased the odds of mortality by 3 times. Platelet and white blood cell diseases more than doubled the odds of mortality. Neoplasms and neurological diseases increased the mortality odds by 93% and 70%, respectively (Table 4).

### 3.5. The Role of Lifestyle, Complications, Comorbidities and Secondary Diagnoses in Mortality

When comparing the lifestyle, complications, comorbidities, and secondary diagnoses of adult patients who were admitted emergently with a primary diagnosis of abscess of anal and rectal regions, differences were noted between those who survived and those who did not. Both deceased elderly and deceased adult patients had higher rates of bacterial infections other than tuberculosis, nonbacterial infections, anemia and, or hemorrhage, respiratory diseases, cardiac diseases, genitourinary system diseases, neurological diseases, fluid and electrolyte disorders, platelet and white blood cell diseases, and skin diseases, as well as trauma, burns, and poisoning. The deceased adults alone had higher rates of hypertension, peripheral vascular diseases, coagulopathy, cerebrovascular disease, liver diseases, endocrine diseases, and neoplasms, as well as alcohol abuse, withdrawal, or dependence (Table 5). Deceased elderly patients alone had higher rates of diabetes and tobacco use. Elderly patients who survived had higher rates of abnormal BMI (underweight, overweight, or obese) compared to those who died (Table 5).

## 4. Discussion

This study’s results demonstrate the differences in mortality rates between adult and elderly patients, along with the main risk factors for mortality in patients admitted emergently with a primary diagnosis of abscess in the anal and rectal regions.

### 4.1. Age as a Risk Factor

Age was seen to be a major risk factor as the mortality rate in elderly patients was more than five times that of adult patients. The association of increasing age with increasing death rates can possibly be attributed to intrinsic factors of aging and more specific anorectal abscess-related factors. In terms of aging, Rosenthal et al. found that the general odds of death in-hospital for all patients increases with each 5-year increase in age [26]. From the age groups of 40–44 to 80–84, the odds of mortality, independent of illness severity, increased from 1.51 to 3.86, respectively [26]. Additionally, between these two age groups, Rosenthal et al. found an increase of mortality in every 5 years of age [26]. Thus, from a broader scope, age is a risk factor for mortality in cases of anorectal abscess diagnosis.

A more specific potential cause of aging in mortality could possibly be due to advances of the disease process from an abscess to NSTI with perineum and retroperitoneal involvement. Increased age enhances the likelihood of these more deleterious types of abscesses in anal or rectal regions, which elevates the chance of mortality in patients [27,28,29]. These abscesses are relatively rarer compared to intraperitoneal abscesses and tend to have a delayed course of action, which leads to delayed diagnoses and improper drainage [9,29]. As aforementioned, delay in diagnosis and timely and complete drainage of abscesses increases risks related to necrotizing tissue [9,12]. Delayed treatment increases mortality in patients with other primary diseases such as hemorrhoids, duodenal ulcers, and ventral hernias [14,15,16]. Even with proper treatment, research has shown these retroperitoneal abscesses can result in an 11–20% mortality rate [28]. These rarer anorectal abscesses are very often seen in males and in the 6th decade of life, which could connect to the relative increase in death rates of elderly patients in this study [28]. Thus, age is a significant risk factor for mortality in patients with anorectal abscesses.

### 4.2. Interplay between HLOS and Operation Status

The next central finding in our study was that operative treatment was a protective factor against mortality, while prolonged hospital length of stay was a risk factor for mortality. In this study, however, those having an operation had a significantly shorter HLOS. Others have reported similar findings. Hsieh et al. found that, in anorectal abscess patients on dialysis, those who had surgery had better in-hospital survival [30]. They also note the possibility that those who received the more conservative non-surgical treatment potentially had more significant comorbidities [30]. Similarly, Dos-Santos et al. found that a predominant cause of death from anorectal abscess (Fournier’s Gangrene) is often associated with comorbidities and results in longer hospital length of stay and greater mortality rates [31].

In our study, it also was observed that patients that had an operation had lower rates of comorbidities compared to those who did not undergo an operation. Therefore, the higher rates of comorbidities in the non-surgical group of patients may have meant they were poor candidates for surgery, and also may have caused them to require a prolonged HLOS and consequently, increased mortality. From a surgical standpoint, irrespective of comorbidities, patients with an abscess need to be operated on as soon as possible, usually in the first 2–3 h, in order to prevent development of NSTI. Having co-morbidities should not be a contraindication for surgical intervention, and in fact should increase the awareness [13]. The common dictum that we should resuscitate patients with NSTI before operating is inappropriate and has deleterious effects. These patients need to be resuscitated simultaneously with surgical intervention and blood and blood products aggressively, and thus the need for resuscitation should not delay surgery. Further, Ramanujam et al. state that more aggressive surgical procedures in abscess treatment reduces complications and the need for further surgery [32]. The reduction in complications and reduced need for further surgery is in line with the aforementioned notion that delayed or improper treatment of abscesses results in worsened outcomes. Lower complication rates may then partially explain both the lowered HLOS and mortality found in the operated group.

Longer hospital length of stay has been shown to be detrimental in many diverse conditions [14,15,16,17,18,19,20,21,22,23,24,25,30,33,34]. Interestingly, for patients with ruptured abdominal aortic aneurysm, longer hospital length of stay was inversely correlated with mortality and thus improved outcomes [35]. In elderly patients with colon cancer that were emergently admitted, there was a non-linear, U-shaped association with HLOS and mortality [36]. Similarly, in patients admitted emergently for *C. Difficile* colitis, hospital length of stay had a J-shaped association with mortality in non-elderly patients and a V-shaped association in elderly patients [37]. In elderly patients with an emergent admission for phlebitis and thrombophlebitis, the association between mortality and HLOS was V-shaped [38]. In patients with acute pancreatitis, HLOS and mortality had a V-shaped association [39].

#### 4.2.1. Other Risk Factors

The other main risk factors of mortality included various pre-existing diseases (specifically respiratory, cardiac, liver, genitourinary, platelet and white blood cell, as well as trauma, burns and poisons, neoplasms, and neurological diseases). Anorectal abscesses can also result in fistulas that lead to NSTI, significantly increasing odds of death [40]. Diabetes is a common comorbidity resulting in mortality by contributing to the development of NSTI [41]. Developing necrotic tissue is often associated with comorbidities that negatively impact one’s immune system, with alcoholism, HIV, and leukemia being the most common contributors [42]; necrotizing perineal infection is uncommon but associated with high mortality in those who are immunocompromised or have diabetes [43]. There is limited research on risk factors associated with perirectal abscess mortality, as most studies on mortality are limited to case studies or studies with low sample sizes. Despite limited research on risk factors for mortality, there is a multitude of research in terms of risk factors for recurrence of perirectal abscess. Akkapulu et al. demonstrate no association between recurrence of abscess and sex, age, and hospital length of stay [44]. Additionally, diabetes and obesity were connected to a significant increase in patients developing anorectal abscesses, but they were not significant in readmission to the hospital for anorectal abscess [4]. Sigmon et al. also mention that common risk factors include smoking and diabetes, along with immunosuppressive drugs and HIV [9].

Because of the aforementioned lack of research into risk factors for mortality from anorectal abscess, this paper is important in its role in addressing some gaps in our understanding. Future research should hopefully explore other pertinent risk factors and attempt to determine the exact mechanism of death by each significant risk factor.

#### 4.2.2. Gender and Abscess Likelihood

Our study showed that there were gender differences in admission to the hospital for patients with a primary diagnosis of peri-anal abscess. For both adult and elderly patients, men were more commonly admitted for peri-anal abscess, which has also been copiously supported by the literature. According to Read et al., males are 1.76 times more likely to have a peri-anal abscess than females [45], while Sigmon et al. reported that males are roughly twice as likely to develop these abscesses as females [9]. Additionally, Sahnan et al. also show a roughly 2:1 male-to-female ratio in abscess frequency [8]. This may be explained by the fact that there are significantly more anal glands in males than females [46].

#### 4.2.3. Differences in Total Hospital Charges 

Both adult and elderly females had longer hospital length of stay than males of their respective age. Interestingly, however, adult females had higher charges compared to adult males while elderly females had no significant difference in charges when compared to elderly males. Owens states that, when analyzing healthcare cost disparities of men and women in different segmental ages (0–18, 18–44, 45–64, and >64), adult females and males aged 45–64 have the largest gap, while those older than 64 have the lowest gap. He attributes this difference to health burdens (osteoporosis, CVD, breast cancer, etc.) due to menopause-associated conditions [47]. Assaf et al. also note that there is a burden of increased healthcare costs related to managing menopausal symptoms, particularly in females aged 45–64 [48]. Thus, complications associated with menopause may have resulted in greater hospital charges in adult women.

#### 4.2.4. Strength of the Study

This study’s strength lies in its patient sample. The patient population is substantial in number and includes a wide array of different hospital types and geographic locations. These aspects of the sample allow generalizability to many different clinical settings across the nation, while also allowing this study to analyze how different hospital characteristics, patient demographics, and patient idiosyncrasies interplay in anorectal abscesses. These strengths allow for future investigations.

#### 4.2.5. Limitations of the Study

This study lacks specifications of abscess location, size, severity, procedure approach, and experience of the provider. Additionally, this study is retrospective: this naturally limits our ability for a cause-and-effect analysis and makes the study prone to bias. Further research on cause of death due to anorectal abscess and case complexity are necessary, especially due to the limited research on anorectal-abscess-associated mortality. An additional limitation is that the NIS database uses ICD codes, which may provide inaccurate disease classifications and restrict potential conclusions. Thus, this paper’s ‘Other risk factors’ section must be qualified in that our ability to interpret interplay with comorbidities may be limited.

## 5. Conclusions

The odds of mortality from anorectal abscesses increased in emergently admitted patients who had non-operative management, preexisting disease, older age, and extended hospital stays.

## Figures and Tables

**Table 1 ijerph-20-05387-t001:** Characteristics of emergently admitted patients with the primary diagnosis of abscess of anal/rectal regions. Data are stratified according to sex categories, NIS 2004–2014.

		Adult, N (%)			Elderly, N (%)		
		Male	Female	*p*	Male	Female	*p*
All Cases		27,646 (69.0%)	12,400 (31.0%)		4463 (63.9%)	2519 (36.1%)	
Race	White	12,232 (51.4%)	5964 (56.4%)	<0.001	2680 (69.9%)	1593 (75.3%)	<0.001
	Black	5340 (22.5%)	2694 (25.5%)		492 (12.8%)	286 (13.5%)	
	Hispanic	4357 (18.3%)	1321 (12.5%)		429 (11.2%)	159 (7.5%)	
	Asian/Pacific Islander	644 (2.7%)	205 (1.9%)		93 (2.4%)	33 (1.6%)	
	Native American	191 (0.8%)	81 (0.8%)		21 (0.5%)	11 (0.5%)	
	Other	1012 (4.3%)	311 (2.9%)		120 (3.1%)	34 (1.6%)	
Income Quartile	Quartile 1	8906 (33.3%)	4168 (34.5%)	<0.001	1306 (29.9%)	729 (29.6%)	<0.001
	Quartile 2	6796 (25.4%)	3116 (25.8%)		1124 (25.7%)	680 (27.6%)	
	Quartile 3	6135 (22.9%)	2657 (22%)		1057 (24.2%)	575 (23.3%)	
	Quartile 4	4943 (18.5%)	2123 (17.6%)		881 (20.2%)	481 (19.5%)	
Insurance	Private Insurance	12,619 (45.8%)	5563 (45%)	0.011	520 (11.7%)	196 (7.8%)	0.670
	Medicare	2834 (10.3%)	1298 (10.5%)		3776 (84.7%)	2254 (89.6%)	
	Medicaid	4188 (15.2%)	2871 (23.2%)		66 (1.5%)	35 (1.4%)	
	Self-Pay	5707 (20.7%)	1915 (15.5%)		37 (0.8%)	9 (0.4%)	
	No Charge	586 (2.1%)	179 (1.4%)		1 (0%)	4 (0.2%)	
	Other	1626 (5.9%)	548 (4.4%)		57 (1.3%)	17 (0.7%)	
Hospital Location	Rural	2815 (10.2%)	1417 (11.5%)	<0.001	624 (14%)	360 (14.3%)	0.400
	Urban: Non-Teaching	11,403 (41.4%)	5062 (41.0%)		2007 (45.1%)	1156 (45.9%)	
	Urban: Teaching	13,299 (48.3%)	5867 (47.5%)		1817 (40.8%)	1000 (39.7%)	
Comorbidities	AIDS	607 (2.2%)	106 (0.9%)	<0.001	6 (0.1%)	0 (0%)	
	Alcohol Abuse	1104 (4%)	159 (1.3%)	<0.001	135 (3%)	11 (0.4%)	<0.001
	Deficiency Anemias	1919 (6.9%)	1551 (12.5%)	<0.001	795 (17.8%)	617 (24.5%)	<0.001
	Rheumatoid Arthritis	225 (0.8%)	332 (2.7%)	<0.001	85 (1.9%)	116 (4.6%)	<0.001
	Chronic Blood Loss	88 (0.3%)	93 (0.8%)	<0.001	58 (1.3%)	28 (1.1%)	0.490
	Congestive Heart Failure	496 (1.8%)	236 (1.9%)	0.450	552 (12.4%)	338 (13.4%)	0.210
	Chronic Pulmonary Disease	2180 (7.9%)	1628 (13.1%)	<0.001	876 (19.6%)	452 (17.9%)	0.090
	Coagulopathy	408 (1.5%)	165 (1.3%)	0.260	172 (3.9%)	63 (2.5%)	0.003
	Depression	1202 (4.3%)	1235 (10.0%)	<0.001	251 (5.6%)	245 (9.7%)	<0.001
	Diabetes, Uncomplicated	5666 (20.5%)	3044 (24.5%)	<0.001	1442 (32.3%)	835 (33.1%)	0.470
	Diabetes, Chronic Complications	770 (2.8%)	546 (4.4%)	<0.001	236 (5.3%)	154 (6.1%)	0.150
	Drug Abuse	1098 (4.0%)	341 (2.8%)	<0.001	14 (0.3%)	6 (0.2%)	0.570
	Hypertension	8174 (29.6%)	3660 (29.5%)	0.920	2913 (65.3%)	1739 (69.0%)	0.001
	Hypothyroidism	499 (1.8%)	829 (6.7%)	<0.001	302 (6.8%)	461 (18.3%)	<0.001
	Liver Disease	655 (2.4%)	203 (1.6%)	<0.001	88 (2%)	40 (1.6%)	0.250
	Lymphoma	146 (0.5%)	45 (0.4%)	0.027	83 (1.9%)	27 (1.1%)	0.011
	Fluid/Electrolyte Disorders	3026 (10.9%)	1907 (15.4%)	<0.001	949 (21.3%)	744 (29.5%)	<0.001
	Metastatic Cancer	346 (1.3%)	160 (1.3%)	0.750	212 (4.8%)	57 (2.3%)	<0.001
	Other Neurological Disorders	586 (2.1%)	343 (2.8%)	<0.001	307 (6.9%)	201 (8%)	0.090
	Obesity	2975 (10.8%)	2345 (18.9%)	<0.001	368 (8.2%)	310 (12.3%)	<0.001
	Paralysis	325 (1.2%)	93 (0.8%)	<0.001	106 (2.4%)	48 (1.9%)	0.200
	Peripheral Vascular Disorders	308 (1.1%)	112 (0.9%)	0.060	344 (7.7%)	132 (5.2%)	<0.001
	Psychoses	706 (2.6%)	488 (3.9%)	<0.001	59 (1.3%)	55 (2.2%)	0.006
	Pulmonary Circulation Disorders	72 (0.3%)	42 (0.3%)	0.170	63 (1.4%)	44 (1.7%)	0.270
	Renal Failure	939 (3.4%)	423 (3.4%)	0.940	629 (14.1%)	332 (13.2%)	0.290
	Solid Tumor	387 (1.4%)	194 (1.6%)	0.200	202 (4.5%)	112 (4.4%)	0.880
	Peptic Ulcer	5 (0%)	0 (0%)	0.330	0 (0%)	0 (0%)	<0.001
	Valvular Disease	230 (0.8%)	136 (1.1%)	0.010	179 (4.0%)	126 (5.0%)	0.052
	Weight Loss	467 (1.7%)	232 (1.9%)	0.200	232 (5.2%)	162 (6.4%)	0.032
Invasive Diagnostic Procedure		3089 (11.2%)	1315 (10.6%)	0.090	532 (11.9%)	263 (10.4%)	0.060
Surgical Procedure		23,494 (85%)	10,404 (83.9%)	0.006	3510 (78.6%)	1856 (73.7%)	<0.001
Invasive or Surgical Procedure		23,810 (86.1%)	10,585 (85.4%)	0.043	3594 (80.5%)	1912 (75.9%)	<0.001
Medical/Surgical Complication Diagnosis		214 (0.8%)	104 (0.8%)	0.500	49 (1.1%)	28 (1.1%)	0.960
Reoperation		249 (0.9%)	100 (0.8%)	0.350	100 (2.2%)	59 (2.3%)	0.790
Deceased		45 (0.2%)	17 (0.1%)	0.550	39 (0.9%)	34 (1.4%)	0.060
		Mean (SD)	Mean (SD)	*p*	Mean (SD)	Mean (SD)	*p*
Age, Years		43.05 (11.69)	41.47 (12.28)	<0.001	73.42 (6.96)	76.08 (8.11)	<0.001
Modified Frailty Index Score		0.72 (0.92)	0.83 (0.99)	<0.001	1.66 (1.12)	1.67 (1.10)	0.580
Time to Invasive Diagnostic Procedure, Days		1.18 (1.89)	1.56 (3.33)	<0.001	2.35 (3.26)	2.74 (3.65)	0.160
Time to Surgical Procedure, Days		0.62 (1.22)	0.69 (1.35)	<0.001	1.12 (5.74)	1.11 (1.77)	0.960
Hospital Length of Stay, Days		3.28 (4.01)	3.64 (4.26)	<0.001	5.02 (6.69)	5.85 (6.02)	<0.001
Total Charges, American Dollars		22,727 (32,984)	23,458 (32,992)	0.043	30,999 (36,644)	31,605 (40,119)	0.530

**Table 2 ijerph-20-05387-t002:** Characteristics of emergently admitted patients with the primary diagnosis of abscess of anal/rectal regions. Data are stratified according to operation status, NIS 2004–2014.

		Adult, N (%)			Elderly, N (%)		
		No Operation	Operation	*p*	No Operation	Operation	*p*
All Cases		6169 (15.3%)	34,137 (84.7%)		1616 (23.1%)	5370 (76.9%)	
Sex, Female		1996 (32.5%)	10,404 (30.7%)	0.006	663 (41.0%)	1856 (34.6%)	<0.001
Race	White	2846 (54.3%)	15,351 (52.7%)	0.002	1010 (73.8%)	3263 (71.2%)	0.002
	Black	1241 (23.7%)	6793 (23.3%)		184 (13.4%)	594 (13.0%)	
	Hispanic	791 (15.1%)	4888 (16.8%)		110 (8.0%)	478 (10.4%)	
	Asian/Pacific Islander	110 (2.1%)	739 (2.5%)		26 (1.9%)	101 (2.2%)	
	Native American	56 (1.1%)	216 (0.7%)		14 (1.0%)	18 (0.4%)	
	Other	202 (3.9%)	1121 (3.9%)		25 (1.8%)	129 (2.8%)	
Income Quartile	Quartile 1	2124 (35.5%)	10,996 (33.2%)	<0.001	477 (30.2%)	1559 (29.7%)	0.960
	Quartile 2	1529 (25.6%)	8425 (25.4%)		415 (26.3%)	1390 (26.4%)	
	Quartile 3	1301 (21.8%)	7548 (22.8%)		377 (23.9%)	1255 (23.9%)	
	Quartile 4	1021 (17.1%)	6153 (18.6%)		309 (19.6%)	1054 (20.0%)	
Insurance	Private Insurance	2508 (40.8%)	15,818 (46.5%)	<0.001	143 (8.9%)	574 (10.7%)	0.220
	Medicare	885 (14.4%)	3254 (9.6%)		1418 (87.9%)	4615 (86.1%)	
	Medicaid	1303 (21.2%)	5793 (17.0%)		26 (1.6%)	75 (1.4%)	
	Self-Pay	1064 (17.3%)	6601 (19.4%)		12 (0.7%)	34 (0.6%)	
	No Charge	93 (1.5%)	672 (2.0%)		0 (0%)	5 (0.1%)	
	Other	293 (4.8%)	1910 (5.6%)		15 (0.9%)	59 (1.1%)	
Hospital Location	Rural	722 (11.8%)	3515 (10.3%)	0.002	262 (16.3%)	722 (13.5%)	0.012
	Urban: Non-Teaching	2546 (41.5%)	14,046 (41.3%)		698 (43.3%)	2468 (46.1%)	
	Urban: Teaching	2868 (46.7%)	16,426 (48.3%)		651 (40.4%)	2167 (40.5%)	
Comorbidities	AIDS	235 (3.8%)	479 (1.4%)	<0.001	2 (0.1%)	4 (0.1%)	0.630
	Alcohol Abuse	239 (3.9%)	1024 (3.0%)	<0.001	29 (1.8%)	117 (2.2%)	0.340
	Deficiency Anemias	810 (13.1%)	2663 (7.8%)	<0.001	374 (23.1%)	1039 (19.3%)	<0.001
	Rheumatoid Arthritis	105 (1.7%)	452 (1.3%)	0.019	47 (2.9%)	154 (2.9%)	0.930
	Chronic Blood Loss	41 (0.7%)	140 (0.4%)	0.006	27 (1.7%)	59 (1.1%)	0.070
	Congestive Heart Failure	161 (2.6%)	571 (1.7%)	<0.001	242 (15.0%)	648 (12.1%)	0.002
	Chronic Pulmonary Disease	669 (10.8%)	3143 (9.2%)	<0.001	308 (19.1%)	1020 (19.0%)	0.950
	Coagulopathy	149 (2.4%)	424 (1.2%)	<0.001	69 (4.3%)	166 (3.1%)	0.021
	Depression	497 (8.1%)	1941 (5.7%)	<0.001	136 (8.4%)	360 (6.7%)	0.019
	Diabetes, Uncomplicated	1440 (23.3%)	7279 (21.3%)	<0.001	503 (31.1%)	1774 (33.0%)	0.150
	Diabetes, Chronic Complications	271 (4.4%)	1045 (3.1%)	<0.001	100 (6.2%)	290 (5.4%)	0.230
	Drug Abuse	310 (5.0%)	1129 (3.3%)	<0.001	7 (0.4%)	13 (0.2%)	0.210
	Hypertension	2044 (33.1%)	9802 (28.7%)	<0.001	1035 (64.0%)	3617 (67.4%)	0.013
	Hypothyroidism	234 (3.8%)	1094 (3.2%)	0.017	174 (10.8%)	590 (11.0%)	0.800
	Liver Disease	168 (2.7%)	691 (2.0%)	<0.001	36 (2.2%)	92 (1.7%)	0.180
	Lymphoma	48 (0.8%)	144 (0.4%)	<0.001	32 (2%)	78 (1.5%)	0.140
	Fluid/Electrolyte Disorders	1049 (17.0%)	3885 (11.4%)	<0.001	442 (27.4%)	1252 (23.3%)	<0.001
	Metastatic Cancer	163 (2.6%)	343 (1.0%)	<0.001	79 (4.9%)	190 (3.5%)	0.013
	Other Neurological Disorders	167 (2.7%)	762 (2.2%)	0.022	145 (9%)	363 (6.8%)	0.003
	Obesity	883 (14.3%)	4443 (13.0%)	0.006	149 (9.2%)	529 (9.9%)	0.450
	Paralysis	126 (2.0%)	292 (0.9%)	<0.001	41 (2.5%)	113 (2.1%)	0.300
	Peripheral Vascular Disorders	94 (1.5%)	326 (1.0%)	<0.001	115 (7.1%)	361 (6.7%)	0.580
	Psychoses	254 (4.1%)	941 (2.8%)	<0.001	32 (2%)	82 (1.5%)	0.210
	Pulmonary Circulation Disorders	24 (0.4%)	90 (0.3%)	0.090	28 (1.7%)	79 (1.5%)	0.450
	Renal Failure	308 (5.0%)	1055 (3.1%)	<0.001	237 (14.7%)	724 (13.5%)	0.230
	Solid Tumor	197 (3.2%)	384 (1.1%)	<0.001	115 (7.1%)	199 (3.7%)	<0.001
	Peptic Ulcer	0 (0%)	5 (0%)	0.340	0 (0%)	0 (0%)	
	Valvular Disease	57 (0.9%)	310 (0.9%)	0.900	75 (4.6%)	230 (4.3%)	0.540
	Weight Loss	192 (3.1%)	507 (1.5%)	<0.001	111 (6.9%)	283 (5.3%)	0.015
Invasive Diagnostic Procedure		499 (8.1%)	3929 (11.5%)	<0.001	140 (8.7%)	655 (12.2%)	<0.001
Medical/Surgical Complication Diagnosis		50 (0.8%)	268 (0.8%)	0.840	15 (0.9%)	62 (1.2%)	0.450
Reoperation		0 (0%)	349 (1.0%)	<0.001	0 (0%)	159 (3.0%)	<0.001
Deceased		20 (0.3%)	42 (0.1%)	<0.001	27 (1.7%)	46 (0.9%)	0.005
		Mean (SD)	Mean (SD)	*p*	Mean (SD)	Mean (SD)	*p*
Age, Years		43.54 (12.02)	42.35 (11.86)	<0.001	75.46 (7.96)	74.06 (7.32)	<0.001
Modified Frailty Index Score		0.91 (1.02)	0.72 (0.92)	<0.001	1.71 (1.11)	1.65 (1.11)	0.052
Time to Invasive Diagnostic Procedure, Days		2.12 (2.23)	1.17 (2.40)	<0.001	3.08 (3.06)	2.34 (3.46)	0.029
Hospital Length of Stay, Days		3.94 (5.33)	3.28 (3.81)	<0.001	5.62 (5.33)	5.23 (6.77)	0.030
Total Charges, American Dollars		22,267 (32,479)	23,062 (32,998)	0.080	27,627 (33,924)	32,303 (38,998)	<0.001

**Table 3 ijerph-20-05387-t003:** Characteristics of emergently admitted patients with the primary diagnosis of abscess of anal/rectal regions. Data are classified according to outcome categories, NIS 2004–2014.

		Adult, N (%)			Elderly, N (%)		
		Survived	Deceased	*p*	Survived	Deceased	*p*
All Cases		40,230 (99.8%)	62 (0.2%)		6910 (99.0%)	73 (1.0%)	
Sex, Female		12,382 (31.0%)	17 (27.4%)	0.550	2483 (36.0%)	34 (46.6%)	0.060
Race	White	18,163 (53.0%)	27 (50.0%)	0.460	4220 (71.7%)	51 (78.5%)	0.730
	Black	8015 (23.4%)	18 (33.3%)		770 (13.1%)	8 (12.3%)	
	Hispanic	5670 (16.5%)	6 (11.1%)		583 (9.9%)	5 (7.7%)	
	Asian/Pacific Islander	847 (2.5%)	2 (3.7%)		127 (2.2%)	0 (0%)	
	Native American	272 (0.8%)	0 (0%)		32 (0.5%)	0 (0%)	
	Other	1321 (3.9%)	1 (1.9%)		153 (2.6%)	1 (1.5%)	
Income Quartile	Quartile 1	13,097 (33.6%)	22 (35.5%)	0.560	2014 (29.8%)	21 (30.0%)	0.880
	Quartile 2	9936 (25.5%)	14 (22.6%)		1785 (26.4%)	20 (28.6%)	
	Quartile 3	8832 (22.6%)	11 (17.7%)		1618 (23.9%)	14 (20.0%)	
	Quartile 4	7156 (18.3%)	15 (24.2%)		1346 (19.9%)	15 (21.4%)	
Insurance	Private Insurance	18,301 (45.6%)	22 (35.5%)	<0.001	713 (10.3%)	3 (4.1%)	0.320
	Medicare	4122 (10.3%)	16 (25.8%)		5961 (86.4%)	70 (95.9%)	
	Medicaid	7075 (17.6%)	16 (25.8%)		101 (1.5%)	0 (0%)	
	Self-Pay	7660 (19.1%)	3 (4.8%)		46 (0.7%)	0 (0%)	
	No Charge	765 (1.9%)	0 (0%)		5 (0.1%)	0 (0%)	
	Other	2195 (5.5%)	5 (8.1%)		74 (1.1%)	0 (0%)	
Hospital Location	Rural	4232 (10.6%)	4 (6.5%)	0.008	974 (14.1%)	9 (12.3%)	0.700
	Urban: Non-Teaching	16,570 (41.4%)	16 (25.8%)		3133 (45.5%)	31 (42.5%)	
	Urban: Teaching	19,245 (48.1%)	42 (67.7%)		2785 (40.4%)	33 (45.2%)	
Comorbidities	AIDS	712 (1.8%)	2 (3.2%)	0.300	5 (0.1%)	1 (1.4%)	0.060
	Alcohol Abuse	1257 (3.1%)	6 (9.7%)	0.003	145 (2.1%)	1 (1.4%)	0.999
	Deficiency Anemias	3459 (8.6%)	11 (17.7%)	0.010	1395 (20.2%)	18 (24.7%)	0.340
	Rheumatoid Arthritis	555 (1.4%)	1 (1.6%)	0.580	199 (2.9%)	2 (2.7%)	0.999
	Chronic Blood Loss	180 (0.4%)	1 (1.6%)	0.240	84 (1.2%)	2 (2.7%)	0.230
	Congestive Heart Failure	721 (1.8%)	11 (17.7%)	<0.001	868 (12.6%)	22 (30.1%)	<0.001
	Chronic Pulmonary Disease	3802 (9.5%)	8 (12.9%)	0.350	1304 (18.9%)	24 (32.9%)	0.002
	Coagulopathy	561 (1.4%)	12 (19.4%)	<0.001	227 (3.3%)	8 (11.0%)	<0.001
	Depression	2434 (6.1%)	3 (4.8%)	0.999	490 (7.1%)	5 (6.8%)	0.940
	Diabetes, Uncomplicated	8700 (21.6%)	16 (25.8%)	0.420	2261 (32.7%)	14 (19.2%)	0.014
	Diabetes, Chronic Complications	1311 (3.3%)	4 (6.5%)	0.140	387 (5.6%)	3 (4.1%)	0.800
	Drug Abuse	1433 (3.6%)	3 (4.8%)	0.490	20 (0.3%)	0 (0%)	0.999
	Hypertension	11,812 (29.4%)	29 (46.8%)	0.003	4601 (66.6%)	50 (68.5%)	0.730
	Hypothyroidism	1324 (3.3%)	4 (6.5%)	0.150	754 (10.9%)	10 (13.7%)	0.450
	Liver Disease	846 (2.1%)	12 (19.4%)	<0.001	126 (1.8%)	2 (2.7%)	0.390
	Lymphoma	188 (0.5%)	3 (4.8%)	0.003	109 (1.6%)	1 (1.4%)	0.999
	Fluid/Electrolyte Disorders	4903 (12.2%)	28 (45.2%)	<0.001	1658 (24%)	35 (47.9%)	<0.001
	Metastatic Cancer	498 (1.2%)	8 (12.9%)	<0.001	261 (3.8%)	8 (11.0%)	0.002
	Other Neurological Disorders	920 (2.3%)	6 (9.7%)	<0.001	495 (7.2%)	13 (17.8%)	<0.001
	Obesity	5318 (13.2%)	8 (12.9%)	0.940	674 (9.8%)	4 (5.5%)	0.320
	Paralysis	417 (1.0%)	0 (0%)	0.999	153 (2.2%)	1 (1.4%)	0.999
	Peripheral Vascular Disorders	415 (1.0%)	5 (8.1%)	<0.001	465 (6.7%)	11 (15.1%)	0.005
	Psychoses	1193 (3.0%)	1 (1.6%)	0.999	113 (1.6%)	1 (1.4%)	0.999
	Pulmonary Circulation Disorders	113 (0.3%)	1 (1.6%)	0.160	103 (1.5%)	4 (5.5%)	0.025
	Renal Failure	1347 (3.3%)	16 (25.8%)	<0.001	933 (13.5%)	28 (38.4%)	<0.001
	Solid Tumor	579 (1.4%)	2 (3.2%)	0.230	308 (4.5%)	6 (8.2%)	0.120
	Peptic Ulcer	5 (0%)	0 (0%)	0.999	0 (0%)	0 (0%)	
	Valvular Disease	366 (0.9%)	1 (1.6%)	0.430	301 (4.4%)	4 (5.5%)	0.560
	Weight Loss	691 (1.7%)	7 (11.3%)	<0.001	373 (5.4%)	21 (28.8%)	<0.001
Invasive Diagnostic Procedure		4414 (11.0%)	12 (19.4%)	0.035	779 (11.3%)	15 (20.5%)	0.013
Surgical Procedure		34,082 (84.7%)	42 (67.7%)	<0.001	5322 (77%)	46 (63.0%)	0.005
Invasive or Surgical Procedure		34,579 (86.0%)	44 (71.0%)	<0.001	5460 (79%)	48 (65.8%)	0.006
Medical/Surgical Complication Diagnosis		315 (0.8%)	3 (4.8%)	0.013	72 (1.0%)	5 (6.8%)	<0.001
Reoperation		339 (0.8%)	9 (14.5%)	<0.001	151 (2.2%)	8 (11.0%)	<0.001
		Mean (SD)	Mean (SD)	*p*	Mean (SD)	Mean (SD)	*p*
Age, Years		42.52 (11.89)	51.42 (10.43)	<0.001	74.33 (7.47)	79.44 (8.75)	<0.001
Modified Frailty Index Score		0.75 (0.94)	1.69 (1.21)	<0.001	1.66 (1.11)	2.29 (1.16)	<0.001
Time to Invasive Diagnostic Procedure, Days		1.28 (2.39)	3.92 (4.38)	0.060	2.38 (3.18)	7.00 (7.88)	0.047
Time to Surgical Procedure, Days		0.64 (1.25)	1.62 (2.82)	0.037	1.10 (4.76)	2.84 (4.81)	0.023
Hospital Length of Stay, Days		3.36 (3.99)	14.60 (15.25)	<0.001	5.25 (6.40)	11.62 (9.35)	<0.001
Total Charges, American Dollars		22,742 (31,511)	159,516 (204,036)	<0.001	30,636 (36,797)	84,490 (80,680)	<0.001

**Table 4 ijerph-20-05387-t004:** Backward logistic regression analysis to evaluate the associations between mortality and different risk factors in patients emergently admitted with a primary diagnosis of abscess of anal and rectal regions (NIS 2005–2014). Mortality was the dependent variable.

Patients’ Characteristics	Mortality	
	N = 47,011	R^2^ = 0.357
	OR (95% CI)	*p*
Number of Events	N = 135	
Age, Years	1.03 (1.02, 1.04)	<0.001
Surgical Procedure	0.49 (0.33, 0.71)	<0.001
Hospital Length of Stay, Days	1.02 (1.01, 1.03)	<0.001
Invasive Procedure	1.55 (0.98, 2.46)	0.060
Respiratory Diseases	4.19 (2.86, 6.13)	<0.001
Cardiac Diseases	3.60 (2.36, 5.50)	<0.001
Liver Diseases	3.62 (2.07, 6.32)	<0.001
Genitourinary System Diseases	3.00 (1.97, 4.56)	<0.001
Platelet and White Blood Cell Diseases	2.43 (1.59, 3.71)	<0.001
Trauma, Burns, and Poisons	7.71 (5.31, 11.19)	<0.001
Neoplasms	1.93 (1.30, 2.87)	0.001
Neurological Diseases	1.70 (1.15, 2.50)	0.008
Sex, Female	Removed Via Stepwise Backward Elimination	
Bacterial Infections (Other than Tuberculosis)		
Coagulopathy		
Peripheral Vascular Diseases		
Fluid and Electrolyte Disorders		
Cerebrovascular Diseases		
Tuberculosis		
Nonbacterial Infections		
Anemia and/or Hemorrhage		
Digestive Diseases other than Liver		
Diabetes		
Drug Abuse/Withdrawal/Dependence		
Alcohol Abuse/Withdrawal/Dependence		
Tobacco Use		
Hypertension		
Endocrine Diseases		
Nutritional/Weight Disorders		
Musculoskeletal System and Connective Tissue Diseases		
Psychiatric Diseases		
Skin Diseases		
Long Term Medication Usage		
Diseases of Oral Cavity, Salivary Glands, and Jaw		
Sleep Disorders		
Lack of Physical Evidence		
Inappropriate Diet and Eating Habits		
High Risk Lifestyle Behaviors		
Social Factors		

**Table 5 ijerph-20-05387-t005:** Secondary diagnoses of patients emergently admitted with a primary diagnosis of abscess of anal and rectal regions (NIS 2004–2014). Data are stratified according to survival status.

	Adult, N (%)			Elderly, N (%)		
Lifestyle, Complications, Comorbidities and Secondary Diagnoses (ICD-9 Codes)	Survived	Deceased	*p* Value	Survived	Deceased	*p* Value
Observations	40,230 (99.8%)	62 (0.2%)		6910 (99%)	73 (1%)	
Tuberculosis (010.0–018.96)	4 (0.0%)	0 (0%)	0.940	0 (0%)	0 (0%)	
Bacterial Infections Other than Tuberculosis (020.0–041.9, 790.7)	7037 (18%)	36 (58%)	<0.001	1849 (27%)	33 (45%)	<0.001
Nonbacterial Infections (042, 795.71, V08, 045.0–139.8, 790.8, and/or presence of Comorbidity of AIDS)	3221 (8%)	12 (19%)	<0.001	342 (5%)	10 (14%)	<0.001
Diabetes (250.0–250.93, V58.67, and/or presence of Comorbidity of Diabetes Uncomplicated or Diabetes Chronic Complications)	10,028 (25%)	20 (32%)	0.180	2654 (38%)	17 (23%)	0.008
Hypertension (401.0–405.99, 796.2, and/or presence of Comorbidity of Hypertension)	11,893 (30%)	29 (47%)	0.003	4611 (67%)	50 (69%)	0.750
Anemia and/or Hemorrhage (280.0–285.9, 784.7, 784.8, and/or presence of Comorbidity of Anemia)	4154 (10%)	20 (32%)	<0.001	1703 (25%)	28 (38%)	0.007
Respiratory Diseases (415.0–417.9, 460–519.9, 784.91, 786, and/or presence of Comorbidity of COPD, ILD or Pulmonary Circulation Disease)	5044 (13%)	37 (60%)	<0.001	1818 (26%)	53 (73%)	<0.001
Coagulopathy (286.0–286.9, 790.92, V58.61, V58.63, and/or presence of Comorbidity of Coagulopathy)	1266 (3%)	13 (21%)	<0.001	791 (11%)	9 (12%)	0.810
Cardiac Diseases (391.X, 392.0, 393.398.99, 410.0–414.9, 420.0–429.9, 794.3X, 785.XX, and/or presence of Comorbidity of CHF or Valvular Diseases)	4383 (11%)	45 (73%)	<0.001	3151 (46%)	55 (75%)	<0.001
Cerebrovascular Diseases (325, 430–438)	200 (0.5%)	4 (7%)	<0.001	277 (4%)	2 (3%)	0.580
Peripheral Vascular Diseases (440–457.9, and/or presence of Comorbidity of Peripheral Vascular Disorders)	2925 (7%)	12 (19%)	<0.001	1024 (15%)	16 (22%)	0.090
Liver Diseases (570–573.9, 790.4, 794.8, and/or presence of Comorbidity of Liver Diseases)	1095 (3%)	14 (23%)	<0.001	167 (2%)	4 (6%)	0.090
Diseases of Digestive System other than Liver (530.00–569.9, 574.0–579.9, 787, 001.0–009.3, and/or presence of Comorbidity of Peptic Ulcer)	12,324 (31%)	25 (40%)	0.100	3025 (44%)	37 (51%)	0.240
Diseases of Oral Cavity, Salivary Glands, and Jaws (520–529)	136 (0.3%)	0 (0%)	0.650	23 (0.3%)	0 (0%)	0.620
Nutritional/Weight Disorders (260–273.9, 275.XX, 277.0–278.8, 783.XX, 799.3–799.4, and/or presence of Comorbidity of Weight Loss)	10,251 (26%)	22 (36%)	0.070	3210 (47%)	33 (45%)	0.830
Endocrine Diseases (240.0–259.9, 991.0–992.9, and/or presence of Comorbidity of Endocrine Diseases)	11,073 (28%)	24 (39%)	0.049	3190 (46%)	26 (36%)	0.070
Genitourinary System Diseases (580.0–629.9, 403.XX, 791.XX, 788.XX, and/or presence of Comorbidity of Renal Diseases)	6733 (17%)	43 (69%)	<0.001	2965 (43%)	59 (81%)	<0.001
Neurological Diseases (317.0–326, 330.0–337.9, 340–359.9, 392, 780.0–780.09, 780.2–780.4, 317–319, 290.XX, 294.XX, 781.0–782.0, and/or presence of Comorbidity of Paralysis or Other Neurological Disorders or Paralysis)	2846 (7%)	18 (29%)	<0.001	1361 (20%)	28 (38%)	<0.001
Diseases of the Musculoskeletal System and Connective Tissue (274.XX, 710.0–739, and/or presence of Comorbidity of Rheumatoid Arthritis or Lupus)	4018 (10%)	10 (16%)	0.110	1730 (25%)	22 (30%)	0.320
Fluid and Electrolyte Disorders (275.0–276.9, 458.0–459.9, and/or presence of Comorbidity of Fluid and Electrolyte Disorders)	5466 (14%)	31 (50%)	<0.001	1871 (27%)	38 (52%)	<0.001
Neoplasms (140.0–239.9, V10.XX, and/or presence of Comorbidity of Lymphoma, Metastatic Diseases, or Tumor)	2785 (7%)	21 (34%)	<0.001	1736 (25%)	25 (34%)	0.070
Platelet and White Blood Cell Diseases (204.0–208.92, 287.0–288.9, 238.71)	2371 (6%)	21 (34%)	<0.001	634 (9%)	14 (19%)	0.003
Psychiatric Diseases (293.XX, 295.0–302.9, 306.0–316, 780.1, V62.8, V15.4, and/or presence of Comorbidity of Psychoses)	4678 (12%)	6 (10%)	0.630	847 (12%)	10 (14%)	0.710
Skin Diseases (680.0–709.9, 782.1–782.9)	5063 (13%)	18 (29%)	<0.001	1206 (18%)	21 (29%)	0.012
Trauma, Burns and Poisoning (800–999)	1592 (4%)	37 (60%)	<0.001	498 (7%)	32 (44%)	<0.001
Drug Abuse/Withdrawal/Dependence (292.0–292.9, 304.0–304.93, 305.2–305.93, and/or presence of Comorbidity of Drug Abuse)	1448 (4%)	3 (5%)	0.600	35 (0.5%)	0 (0%)	0.540
Alcohol Abuse/Withdrawal/Dependence (291.0–291.9, 303.0–303.93, 305.0–305.03, and/or presence of Comorbidity of Alcohol Abuse)	1257 (3%)	6 (10%)	0.003	145 (2%)	1 (1%)	0.670
Tobacco Use (305.1)	12,035 (30%)	13 (21%)	0.120	1321 (19%)	6 (8%)	0.018
Long-Term Medications/Radiotherapy (V58.0–V58-2, V58.62, V58.64–V58.66, V58.68–V58.69)	1881 (5%)	6 (10%)	0.060	702 (10%)	5 (7%)	0.350
Social Factors (V60.0–V62.6, V63.0–V64.3, V15.81)	1742 (4%)	1 (2%)	0.290	138 (2%)	2 (3%)	0.650
Sleep Disorders (327, 780.5, V69.4, V69.5)	1498 (4%)	4 (7%)	0.260	316 (5%)	2 (3%)	0.460
Lack of Physical Exercise (V69.0)	0 (0)	0 (0%)		0 (0%)	0 (0%)	
Inappropriate Diet and Eating Habits (V69.1)	0 (0)	0 (0%)		0 (0%)	0 (0%)	
High Risk Lifestyle Behaviors (V69.2, V69.3)	5 (0.0)	0 (0%)	0.930	0 (0%)	0 (0%)	
Body Mass Index of Less than 18.9 (V85.0)	87 (4%)	0 (0%)	0.410	43 (13%)	0 (0%)	0.020
Body Mass Index of 19–24.9 (V85.1)	105 (5%)	1 (20%)		5 (10%)	2 (67%)	
Body Mass Index of 25.0–29.9 (V85.21–V85.25)	157 (7%)	0 (0%)		41 (12%)	0 (0%)	
Body Mass Index of 30.0 and over (V85.30–V85.45)	1785 (84%)	4 (80%)		217 (65%)	1 (33%)	

## Data Availability

Data will be available upon request.

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
