# Peer review of "Elderly Patients Managed Non-Operatively with Abscesses of the Anorectal Region Have Five Times Higher Rate of Mortality Compared to Non-Elderly"

_ijerph, 2023, doi:10.3390/ijerph20075387_

Round 1

Reviewer 1 Report

Dear authors,

Thank you for the opportunity to review the paper. It required a lot of work and dedication to analyze this amount of data. However, in my opinion, the study does not provide enough new insight into the analyzed disease to be published in the present form and it contains some design flaws.

This is a retrospective analysis of a national database to investigate the mortality risk factors in patients with the primary diagnosis of anorectal abscess. The conclusion is that operative management lowers the risk of mortality, and that mortality increases with age. This is commonly known, and therefore the study does not add to the medical knowledge. The authors state that the literature on mortality in anorectal abscesses is scarce. There is an abundance of literature on anorectal abscesses mortality in Crohn’s disease, cancer patients, immunosuppressed patients, etc. The anorectal abscess by itself, with no additional risk factors and proper treatment, is not lethal, which may be the reason for the lack of literature. When the treatment is delayed or wrong, the fistulas form, which also is described in a multitude of papers. The authors themselves on multiple occasions elaborate on NSTI, which is a complication of anorectal abscess, even though the analyzed database does not provide information on NSTI occurrence.

The national database used for the study provides a large patient sample, but it must be considered that being based on the ICD codes it does not provide accurate data and it must be clearly stated in the study and taken into account when forming conclusions.

E.g. the extrapolation of the functional status based on the comorbidities in the study is controversial, as the diagnoses in the database are often vague. A patient with diagnosed “coagulopathy” or “weight loss” can very well be fully independent. “Other neurological disorders”, “solid tumor”, “medical or surgical complications” etc. could be various diseases, causing various level of dependence and having various impact on mortality.

I suggest using only data which is unambiguous (e.g. HLOS, cost of hospital stay, type of hospital, demographic data, surgical and invasive intervention, BMI) for analysis and other data (the comorbidities) should be used cautiously and with a proper discussion of the weaknesses of the study.

Below are some other comments:

Line 73-82 characterizing the advantages of the database would be more appropriate in the discussion section, in my opinion. The disadvantages of the database should be mentioned, too.

Line 214: It cannot be assumed that the patients who died, died of NSTI. There could be multiple reasons for death, also arising from comorbidities and not necessarily from the abscess.

Discussion:

Lines 273-284: There is no evidence whatsoever that in the analyzed group of patients, the NSTI was present and/ or caused death. NSTI as a complication of anorectal abscesses and a potential cause of mortality may be mentioned, but it should be clearly stated that this study cannot determine its role.

Line 306 and 337-343: Again, the authors elaborate on NSTI, while the data of the study does not include any information on NSTI.

Line 317: did you mean “undergoing colorectal cancer surgery”? It seems something is missing in this sentence

Line 345-352: the authors state that there is limited research on mortality in anorectal abscesses and therefore dedicate the paragraph to the topic of anorectal abscess recurrence. This is irrelevant to the study.

Paragraph on HLOS: The long listing of other conditions and the impact of HLOS on them is not relevant to the topic. I suggest leaving it at “Longer hospital length of stay has been shown to be detrimental in many conditions” with a reference list only.

In the results section, the authors state that there were differences in invasive and surgical intervention time between genders, as well as a difference in charges in the adult patients(Line 149-150: “Adult females had longer hospital length of stay (HLOS), longer time to invasive diagnostic procedures and longer time to surgical procedures than adult males. Adult females also were charged more money than adult males”). Some of these differences disappeared in the elderly group. It would be interesting to discuss the possible reasons for these differences. Also, the abnormal BMI in elderly patients who survived is a surprising result worth discussing.

Conclusions: “The odds of mortality from anorectal abscesses increased in emergently admitted patients who were managed non-operatively and subsequently had extended hospital stays and pre-existing disease.”  - this is not accurate. It omits the impact of age, which is even mentioned in the title of the manuscript. The study does not prove that the prolonged hospital stay was caused by non-operative management, therefore this conclusion is not supported by the results.  Also, the current sentence sounds like pre-existing diseases are a consequence of non-operative management, which they obviously are not.

Author Response

Please see attachment with responses to you. Thank you. 

Reviewer 2 Report

dear authors I congratulate you on the study. A good number of cases is included, and the data well analyzed.

Author Response

Please see attachment with responses to all. Thank you!

Reviewer 3 Report

First of all, I want to congratulate the authors for their work.

Here are my comments:

1)This is a retrospective study and has many limitations. Risk factors for mortality  in patients with anorectal abscess are been discussed previous in many studies. This study has strength by large patient population but there are many important factors for mortality not assessed by the authors: abscess location, type of procedure, sepsis or MSOF at the admission in the hospital etc.

2) page 3 line 48 I&D is not previously mentioned in the text.

Author Response

Hello, please see attachment with responses to all. Thank you!
